# Photoluminescence of ZnO Nanowires: A Review

**DOI:** 10.3390/nano10050857

**Published:** 2020-04-29

**Authors:** Andres Galdámez-Martinez, Guillermo Santana, Frank Güell, Paulina R. Martínez-Alanis, Ateet Dutt

**Affiliations:** 1Instituto de Investigaciones en Materiales, Universidad Nacional Autónoma de México, Coyoacan, México City 04510, Mexico; galdamez.m@ciencias.unam.mx; 2ENFOCAT-IN^2^UB, Universitat de Barcelona, C/Martí i Franquès 1, 08028 Barcelona, Catalunya, Spain; frank@el.ub.edu (F.G.); pmartinez@el.ub.edu (P.R.M.-A.)

**Keywords:** ZnO nanowires, VLS, hydrothermal, photoluminescence, emission mechanism

## Abstract

One-dimensional ZnO nanostructures (nanowires/nanorods) are attractive materials for applications such as gas sensors, biosensors, solar cells, and photocatalysts. This is due to the relatively easy production process of these kinds of nanostructures with excellent charge carrier transport properties and high crystalline quality. In this work, we review the photoluminescence (PL) properties of single and collective ZnO nanowires and nanorods. As different growth techniques were obtained for the presented samples, a brief review of two popular growth methods, vapor-liquid-solid (VLS) and hydrothermal, is shown. Then, a discussion of the emission process and characteristics of the near-band edge excitonic emission (NBE) and deep-level emission (DLE) bands is presented. Their respective contribution to the total emission of the nanostructure is discussed using the spatial information distribution obtained by scanning transmission electron microscopy−cathodoluminescence (STEM-CL) measurements. Also, the influence of surface effects on the photoluminescence of ZnO nanowires, as well as the temperature dependence, is briefly discussed for both ultraviolet and visible emissions. Finally, we present a discussion of the size reduction effects of the two main photoluminescent bands of ZnO. For a wide emission (near ultra-violet and visible), which has sometimes been attributed to different origins, we present a summary of the different native point defects or trap centers in ZnO as a cause for the different deep-level emission bands.

## 1. Introduction

Similar to many binary compounds, zinc oxide (ZnO) has been widely used in the last century in a large number of daily applications such as cosmetics (UV screening), antibacterial applications, pigments, catalysts, piezoelectric transducers, and so on [1]. However, as the scientific community has focused so much attention on the synthesis and applications of semiconductor nanostructures, in the last few decades, ZnO nanostructures have gained much popularity due to their belonging to the rich family of possible nanostructures with tunable morphology and properties [2]. 

ZnO is a II-VI oxide semiconductor with a strong ionic character of chemical bonds [3]. Under ambient conditions, ZnO has a wurtzite crystalline structure with hexagonal unit cell and two lattice parameters, a and c with c/a ratio between 1.5393 and 1.6035 that is transformed into a metastable rock salt structure at high pressures [4]. With an exciton binding energy of 60 meV and wide direct bandgap of 3.4 eV at room temperature [5], ZnO can be considered an alternative to GaN for the construction of short-wavelength optoelectronic devices such as light-emitting diodes, UV lasers, and UV detectors, among other [6]. 

ZnO presents several advantages over GaN in these kinds of applications, not only because of its lower production cost but also its excellent radiation resistance [7], high electron mobility [8], greater exciton binding energy, and the ability to grow crystal structures. Nonetheless, the high performance and much more mature technology based on GaN puts severe limitations on the commercial use of ZnO in these applications.

Also, ZnO is a unique material that, besides its optoelectronic semiconductor behavior, exhibits exciting properties such as piezoelectricity [9] and pyroelectricity [10], which further make it a versatile, functional material. 

As mentioned before, a wide diversity of different nanostructure morphologies can be achieved with ZnO using chemical or physical methods including thermal evaporation [11], chemical vapor deposition (CVD) [12,13], sol-gel method [14], hydrothermal [15], molecular beam epitaxy (MBE) [16], metalorganic vapor-phase epitaxy (MOVPE) [17], and solvothermal growth [18]. Among the most reported ZnO nanostructures are zero dimensional systems 0D (nanoparticles NPs [19]), one-dimensional (1D; nanowires NWs [20], nanorods NRs [21], nanotubes NTs [22]), two-dimensional (2D; nanobelts [23], thin films [24]), and more complex morphologies such as nanohelix [25], nanorings [26], nanocages [27], nanoflowers [28], or comb-like structures [29].

Among the above-mentioned nanostructures, nanowires and nanorods represent a significant and broad class of 1D nanostructures and, therefore, are at the forefront of nanoscience and nanotechnology. NWs and NRs can present a polycrystalline or monocrystalline structure and highly anisotropic properties, which could result in rapid growth in one particular direction [20,21]. NWs typically have a cylindrical, hexagonal, square, or triangular cross-section, which is much smaller than the overall length of these 1D nanostructures.

Such ZnO nanostructures have attracted much attention due to excellent charge carrier transport properties and high crystalline quality, which are ideal for optoelectronic and magnetic applications. Recent work on ZnO has demonstrated ferromagnetism properties by doping with transition metals for spintronic devices [30]. Also, detection and photocatalytic applications have been extensively studied due to the large surface area and volume ratio and the ability to adjust the bandgap through substitution with suitable dopants [31,32].

As applications of 1D ZnO nanostructures often rely on the optoelectronic properties of the system, a deep understanding of the mechanisms of light interaction (which are heavily influenced by the lattice dynamics of the system [33]) is desired.

As mentioned before, the literature on ZnO 1D nanostructures is vast; therefore, there are many reviews describing the different synthesis methods [34,35], optical [36] and electrical [37] properties, and practical applications of ZnO nanowires [38,39] and nanorods [40], including photocatalyst material [41], sensing platforms (chemical [42] and biological [43,44]), antibacterial element [45], light emissions sources(LEDs [46] and lasers [47]), dye-sensitized or thin film solar cells [48], field effect transistors [49], and piezoelectric nanogenerators [50].

Furthermore, their non-toxic nature makes ZnO nanowires a good candidate for biological applications such as urea, glucose, or DNA biosensors [44] and electric cardiac biomarker detection [43]. Also, the reported antibacterial activities of ZnO nanostructures have inspired a wide range of antimicrobial applications.

However, to our best knowledge, review reports are lacking where the photoluminescence properties and emission mechanism of ZnO nanowires have been discussed. For that reason, this paper presents an extensive review of the latest research of the luminescence properties of ZnO nanowires. A discussion of the sometimes controversial or contradictory light-emitting mechanisms in both near-band edge excitonic emission (NBE) and deep-level emission (DLE) bands is made as well as their respective contribution to nanoscale systems. Lastly, a discussion of the size reduction effects over the two main photoluminescent bands of ZnO is presented. For a broad emission (near UV and visible), which has sometimes been attributed to different origins, we present a summary of the different native point defects or trap centers in ZnO as a cause for the different deep-level emission bands.

## 2. ZnO Structure and Main Growth Techniques

### 2.1. Crystalline Structure of ZnO

Under ambient pressure and temperature, ZnO crystallizes in the hexagonal wurtzite structure (Figure 1), where each Zn^2+^ ion is surrounded by four O^2−^ ions in tetrahedral coordination (an indicator of the presence of a covalent sp^3^ bond) and vice versa. This hexagonal structure belongs to the C_6V_^4^ class (Schoenflies notation) and the space group P6_3_mc (Hermann–Mauguin notation). The spatial distribution of the interpenetrated Zn^2+^ and O^2−^ tetrahedrons and the mixed ionic-covalent character of the Zn-O bond (with nearly equal contribution from each bonding type) are responsible for the intrinsic polarity in the ZnO crystal along the *c*-axis.

This polarity is responsible for the piezoelectric and spontaneous polarization properties of ZnO and is also a critical factor in crystalline growth and defect generation [41]. This feature gives rise to some interesting chemical reactions that are possible with this semiconductor oxide.

The ZnO hexagonal lattice has four characteristic crystal faces, and the c-axis orientated ones are the Zn-terminated [0001] and the O-terminated $\left[000\bar{1}\right]$, both exhibiting a polar character. The non-polar $\left[11\bar{2}0\right]$ and $\left[10\bar{1}0\right]$ are a-axis oriented and have the same number of Zn and O atoms [51]. Despite the inherent instability of the polar faces, the most common crystal orientations of the ZnO structure correspond to the [0001] and $\left[000\bar{1}\right]$ faces; therefore, additional positive or negative charges are required to stabilize the surface [28].

The lattice parameters determined at room temperature by X-ray diffraction (XRD) are *a* = 3.25 Å and *c* = 5.20 Å with the ratio *c*/*a* = 1.6 in a perfect wurtzite structure. The values of the *a*-parameter usually range from 3.2475 to 3.2501 Å and from 5.2042 to 5.2075 Å for the *c*-parameter. The variations in the lattice parameters are the result of doping, defects, strain, or temperature effects [52].

On the other hand, at high pressures, a halite (rock salt cubic lattice) crystal structure can be achieved. In this phase, the ZnO exhibits an indirect semiconductor character with a bandgap of 2.7 eV [53]. 

### 2.2. ZnO Nanowires

One-dimensional semiconductor nanostructures such as nanowires or nanorods have received significant attention due to their versatile set of applications, among which nanoelectronic devices, photovoltaic systems, and biosensors are found. Furthermore, the valuable properties of these kinds of nanomaterials, including a direct bandgap and large exciton binding energy, make them a promising candidate for the fabrication of efficient optoelectronic nanodevices.

Several chemical and physical methods have been used for the synthesis of ZnO nanowires. In most of the reported methodologies, a bottom-up approach is preferred due to better control in the morphological parameters such as the aspect ratio, orientation, defects, surface density, and the compositional flexibility of obtaining heterostructures [4]. These properties have a significant influence over the efficiency and performance of the diverse applications of the nanowires. Among the most used growth methods are physical deposition techniques such as sputtering, atomic layer deposition, or chemical ones as spray pyrolysis, sol-gel, electrochemical deposition, hydrothermal technology, or all-inkjet-printing approach. 

The linear growth of ZnO NWs on Zn metal substrates is also quite promising due to its ease and control over the growth process [54]. A more detailed exposition of these different growth techniques can be found in the work of Rong et al. [35]. However, among these techniques, the vapor-liquid-solid technique (VLS) and hydrothermal technique are the most commonly used bottom-up ZnO nanowires growth processes [49]. These two techniques are discussed below. 

### 2.3. Vapor-Liquid-Solid Technique

In a CVD process, the precursors are applied to the heated surface of a substrate where specific reactions take place for the formation of the desired material. For the synthesis of nanostructures, the process is performed on a substrate with previously deposited catalyst metal nanoparticles. 

Accordingly, due to the phases presented in the system, the growth techniques can be divided into two groups: vapor-liquid-solid (VLS) and vapor-solid-solid (VSS) mechanisms [39]. 

In the early 1960s, Wagner and Ellis developed the VLS technique to obtain micrometric whiskers. A standard VLS process starts with the diffusion of the reagent gas species in a nanometric “liquid” metal particle (e.g., Au, Cu, Ni, and Sn are employed as catalyst metals [55]) forming a eutectic with the material for the formation of the nanowire. The continuous absorption of the gaseous reagents into the catalyst droplet contributes to a super-saturated solution resulting in the nucleation, crystallization, and growth of a single nanowire from each drop (Figure 2a,b). 

By changing the gaseous reagents in the system, it is possible to obtain both radial or axial heterostructures, which makes this technique quite flexible [56]. The first ZnO nanowires with perfect vertical alignment to the substrate were reported by Huang [57], where a laser emission of the system was also found under UV at room temperature. However, to obtain ZnO nanowires with proper alignment, a low growth rate (5.5 nm/s) regime is required [58].

The alignment of the ZnO nanowires is strongly related to the epitaxial relationship between ZnO and the substrate. Small variations between the ZnO crystallographic network and some materials have allowed NWs to be obtained with excellent alignment on sapphire substrates [57] and several nitrides (GaN, AlGaN, AlN) [59].

**Figure 2 nanomaterials-10-00857-f002:**
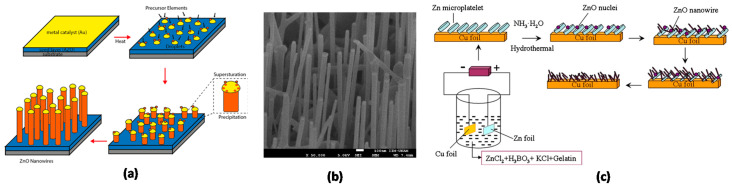
(**a**) Basic principle of the vapor-liquid-solid (VLS) growth mechanism of ZnO NWs using gold as a metal catalyst. (**b**) 45° side-view SEM image of hexagonally ordered ZnO nanorod grown by VLS method on patterned Au-covered substrates. (**c**) Schematic representation of the growth of nanowires by the hydrothermal technique [60]. Copyright 2015 Elsevier.

Even though the growth process in the VLS technique is relatively simple, the high number of experimental parameters makes the method non-trivial. The source temperature, substrate temperature, growth time, carrier gas concentration and flow, the type of substrate, and the characteristics of the catalyst are among the leading parameters to control for precise growth of ZnO nanowires [61]. In particular, the total chamber pressure, partial oxygen pressure, catalyst thickness, and the growth temperature are the most critical parameters of the process [4]. 

### 2.4. Hydrothermal Technique

Although the VLS growth technique allows obtaining nanostructures with an excellent crystalline structure, the high temperatures at which the growth must be performed present a disadvantage when using organic substrates, which could lead to the manufacture of flexible devices. Due to this, different methods of fabrication of ZnO NWs from solutions have been investigated [62]. ZnO, as well as ordinary divalent metals, can crystallize through the hydrolysis of metal precursor salts (e.g., ZnAc dihydrate, zinc nitrate, among others) in a basic solution (e.g.,potassium hydroxide (KOH )and sodium hydroxide (NaOH)). The temperature and pH of the solution determine the formation of tetrahedrally coordinated Zn^2+^ ions, whose dehydration (by proton transfer) will eventually lead to the growth of ZnO nanostructure. The minimization of the energy of the system is considered the driving force for crystalline growth, while the reversible equilibrium determines the chemical reactions in the aqueous system. The (0001) ZnO polar surface has a strong affinity to attract and absorb ions, creating opposite Zn and O terminal surfaces, which lead to the anisotropic growth of the nanowires.

Since chemical reactions take place in an aqueous medium, the term hydrothermal is frequently used to describe the process. 

The typical growth stages of the hydrothermal process are shown in Figure 2c. First, a seed layer of ZnO nanoparticles must be deposited on the substrate. To this end, a thin film of ZnO commonly deposited by spin coating is subdued to a thermal decomposition (>200 °C) to form the ZnO islands. This seed layer facilitates the nucleation of ZnO nanowires by decreasing the thermodynamic barrier for the formation. In addition to the already mentioned nutrient solution (alkaline solution containing a Zn^2+^), a guiding agent (e.g., polyethyleneimine (PEI)) is used in order to decrease the lateral growth of the NWs and maximize their length. The substrate with the ZnO seed layer is kept in the nutrient solution at a constant temperature to ensure stable growth regimes. Finally, heat treatment removes the organic residues from the ZnO surface. 

In the same way as for the VLS method, in order to obtain NW arrangements with controlled morphology, orientation, surface density, and aspect ratio, the parameters involved in growth must be optimized. Some of the parameters affecting the morphology of the NWs are the reagents, seed layer characteristics, growth time, bath temperature, pH of the solution, guiding agents, and mechanical agitation [4].

A summary of different synthesis methods of 1D ZnO nanostructures and reported photoluminescent centers is compared and presented in Table 1. In this table, we present some of the most relevant works on the photoluminescence properties of ZnO nanowires in recent years. Some older works are also included due to their relevance in the area and for a better comparison. 

## 3. Properties of ZnO

### 3.1. Optoelectronic Properties of ZnO

Broadband metal oxide semiconductors have characteristic properties, such as larger bandgap, better electron mobility, and proper light absorption. As a result, they are suitable for high temperature and power electronic devices and short UV optoelectronics. Within this family of broadband semiconductors, ZnO is found as an appropriate material with many favorable properties for these kinds of applications. Also, its excellent resistance to radiation makes it a promising candidate for space and nuclear applications. 

The direct valence band of the wurtzite ZnO structure is split into three states Figure 3a, commonly called A (also called the light hole band), B (also called the heavy hole band), and C (also called the crystal-field split-band) sub-bands, due to crystal field and spin-orbit interactions [63].

**Table 1 nanomaterials-10-00857-t001:** 1D ZnO nanostructures: morphology, methods of preparation, size, and photoluminescence emission.

ZnO Nanostructure	Synthesis Method	Growth Reagents	Substrate	Characteristic Size	PL (eV)	
					NBE	DLE
Randomly orientated NWs [64]	Catalyst-free chemical vapor deposition (CVD)	ZnO	Graphene quantum dots	d = 10–30 nm	3.26 Free exciton	Centered 2.48 O vacancy Zn interstitial O interstitial
				_		
Nanowire [65]	Chemical vapor deposition	Metallic zinc	Silicon	d = 50–200 nm	3.263 Near-band edge electron transition	1.8, 2.2 and 2.5 O vacancies Zn vacancies
				l = 6–8 μm		
Nanowire [66]	Preheating hydrothermal	Zinc nitrate hexahydrate	Glass coated with ZnO thin film	d = 60–80 nm	3.36–3.27 D^0^X, A^0^X, two- electron satellite (TES), donor-acceptor pairs (DAP)	1.5–2.5 Absorbed hydroxyl group
				l = 10 μm		
Nanowire [15]	Low temperature hydrothermal	Zinc chloride, Zinc nitrate hexahydrate, Zinc acetate dihydrate	-	d = 43–62 nm	3.298	2.066-1.937
				l = 1.1–3.3 μm		
Nanowire [67]	Chemical vapor transport (CVT)	Zinc gas precursor Gold as catalyst	*α* sapphire	d = 100 nm	3.263	2.48
				l = 15–20 μm		
Nanowire [68]	Vapor-liquid-solid	ZnO, graphite	Quartz, sapphire, and SiO_2_/Si substrates	d = 40–200 nm	3.38-3.33	2.81-1.55 O vacancy, Au impurities, Zn interstitial
				l = 0.2–2.0 μm		
Nanowire [69]	Electrochemically	Zinc nitrate	Zinc crystal	d = 170 nm	3.17	2.43 2.17 2.00
				l = 2.3 μm		
Nanorods [70]	Vapor phase technique	Zn powder	glass substrates	d = 50 nm	3.25 band-to-band transitions bound excitons	2.5 Zn vacancies
				l = 0.8–1.0 μm		
Nanowire [71]	Hydrothermal method	Ammonium persulphate, sodium hydroxide	Zn foil	d = 10–250 nm	3.263	2.851 Electronic defect
				l > 10 μm		
Nanowire [72]	Reactive magnetron sputtering	ZnO target	Paper and PET substrates	d = 20–40 nm	3.36 D^0^X	2.30
				-		
Nanowire [61,73]	Vapor–liquid–solid	ZnO; graphite	AZO film	d = 30–70 nm l = 7 μm	3.28- 3.25, 3.09	2.49-2.13 O vacancy Zn vacancy
Nanowire [74]	Catalyzed vapor–liquid–solid (VLS) using pulsed laser	ZnO ceramic target	Au-coated Al_2_O_3_ (0001)	d = 70 nm l = 8 μm	3.263	2.38 Native point defect
Nanowire [75]	Hydrothermal method	Zinc acetate dihydrate	ITO coated glass	d = 90 nm l = 630 nm	3.263	2.361-1.984 O vacancy
Nanowire [76]	Plasma-assisted molecular beam epitaxy (PAMBE)	Oxygen plasma and Zn vapor	c-plane sapphire (0001) substrate	d = 50–100 nm	3.4	2.1 O vacancy 1.8 Zn interstitial
				l = 0.5–2.0 μm		

Flower-like nano [77]	Ultrasonic assisted hydrothermal method	Zinc acetate dihydrate	-	nano-scale diameter micro-scale length	-	3.02 Zn vacancy 2.33 O vacancy 2.19 O interstitial
Nanowire [78]	Vapor–solid Vapor–liquid–solid	ZnO, graphite	Silicon wafers	d= 15–80 nm	3.35	2.34, 2.57 O and Zn vacancies, ionized Zn interstitial
Nanowire [79]	Metal–vapor deposition method	Metal zinc powder	Alumina substrate	l several μm	3.26 Free exciton	2.44 singly ionized oxygen vacancy
				d = 55 nm		

However, it is possible to make measurements of the excitonic band of the material with a resolution of 40 μeV; the symmetry associated with the sub-band *A* (*Г*_9_ or *Г*_7_) remained an object of debate in the scientific community for more than 50 years [33]. However, recent studies of the magneto-optic exciton fine structure made by Ding et al. confirmed the *Г*_7_ the symmetry of the valence band according to the unique selection rules of Zeeman splitting lines [80]. There have been some comprehensive studies to establish the refractive index and dielectric constant of ZnO [81,82,83]. In the work of Yoshikawa et al. [81], the high frequency and static dielectric constants were measured using spectroscopic ellipsometry. The measurements were carried out on the surface parallel to the c axis, which allowed them to determine the perpendicular (*E*⊥*c*) and parallel (*E*||*c*) components of the optical constants. The values for the dielectric constants are *ε*_∞_ = 3.68 and *ε*_s_ = 7.40 for *E*⊥*c* (polarization perpendicular to *c*-axis) and *ε*_∞_ = 3.72 and *ε*_s_ = 8.49for *E*||*c* (parallel to c-axis). In the work of Gencyilmaz, the refractive index and extinction coefficient for the wurtzite structure of ZnO were calculated by using the envelope method, and the results are shown in Figure 3b [84]. The electron mobility in a single ZnO crystal measured by Hall effect at room temperature is 239 cm^2^ V^−1^s^−1^ and exhibits an *N*-type conductivity [85]. Characteristic photoluminescence of n-type bulk ZnO is shown (Figure 3c), measured at 4.2 K using a 325 nm He-Cd laser [33]. The emission of the material spans the UV to the green/orange spectral range. In the spectrum, the lines originated from bound exciton (BE), longitudinal optical (LO) phonon replicas, and also, donor-acceptor-pair (DAP) transitions are presented in the UV region. Also, the defect-induced emission band centered around 2.45 eV extending from the blue to the green range is presented. 

**Figure 3 nanomaterials-10-00857-f003:**
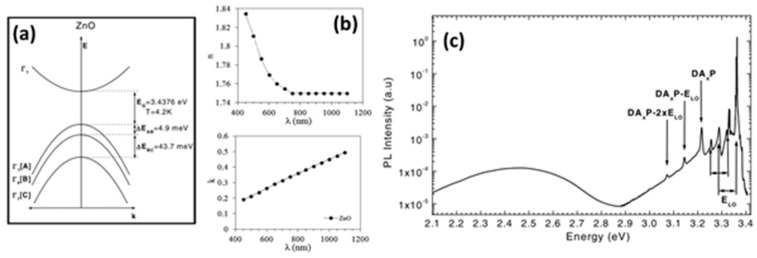
(**a**) Band structure splitting of the hexagonal ZnO structure caused by the crystal field and spin-orbit splitting [63]. Copyright 2004 John Wiley and Sons (**b**) Refractive index (*n*) and extinction coefficient (*k*) of bulk ZnO obtained by the envelope method [84]. Copyright 2012 AIP Publishing. (**c**) Photoluminescence spectrum of bulk ZnO obtained with He-Cd excitation. Excitonic, donor-acceptor pair (DA*_x_*P), and longitudinal optical phonon replicas (LO) are indicated [33]. Copyright 2004 John Wiley and Sons.

### 3.2. Photoluminescence of ZnO Nanowires

Since the synthesis of crystals of ZnO can be carried out by different methodologies, obtained systems can be from tens of nanometers to a few millimeters; therefore, the optical properties can vary enormously.

The luminescence properties of single and collective ZnO nanostructures have been intensively studied using photoluminescence (PL) spectroscopy. The typical photoluminescence spectrum of the nanostructured zinc oxide (ZnO) shows its band edge and exciton luminescence in the UV region. It also presents a green-centered broadband (GL) commonly related to deep-level defects [46]. The first of these bands is usually reported in the literature as NBE (near-band edge excitonic), while the second is known as the DLE band (deep-level emission) [86]. The sizeable binding exciton energy of the ZnO (60 meV, nearly three times that of gallium nitride [87]) is responsible for the room temperature luminescence of the material and that it persists at temperatures as high as 700 K [88]. Usually, the ratio between the integrated spectral intensity of NBE and DLE bands is used to estimate the contribution of recombination due to defect levels [61].

Nonetheless, in ZnO nanostructures, the interaction between these bands becomes more complex, where the high surface-volume ratios result in an increase in surface defects, which could finally affect the processes of photoluminescence emission. 

#### 3.2.1. Near-Band Edge Excitonic Emission (NBE)

ZnO is a bright emitter in the UV region, so it is not hard to wonder why there are numerous photoluminescence investigations of the near-band gap recombinations. The ZnO UV emission (also called near-band emission, NBE) is located close to its absorption edge (3.35 eV [33]) and is produced by excitonic or band-band recombinations. Bound excitons are extrinsic transitions and are related to dopants, native defects, or complexes, which usually create discrete electronic states in the bandgap [52] and at low temperatures are the dominant radiative channels [51]. When the ZnO structure has high crystallinity, the UV emission is more intense than the emission in the visible region [89].

In the work of Qiu et al. [66], a detailed study of the near-band excitonic emission of ultralong ZnO nanowires is presented. The nanostructures were synthesized using a hydrothermal preheating technique, obtaining nanowires as long as 10 μm with a diameter varying from 50 to 80 nm. In Figure 4, the position of the different exciton transitions reported in [66] is schematized.

Although it is known that the NBE emission peak is composed of several peaks and shoulders of less intensity, in many works, its study is carried out indistinctly, analyzing only the sum of these emissions.

However, some studies focus on high-resolution photoluminescence in the area of output emission in detail due to the high consistency that exists between the measured values with respect to those previously reported [33,66,72,90,91,92,93,94].

In the 6K spectra reported by Qiu et al. [66], three peaks can be observed at 3.363, 3.303, and 3.228 eV, and another four shoulders at 3.367, 3.357, 3.353, and 3.342 eV. In high bandgap materials such as ZnO, excitons can be bound to neutral or charged donors and acceptors, due to the strong coupling between them [52]. The central peak at 3.363 and its two lateral shoulders (3.367 and 3.357) are tentatively ascribed (based in the previous work of Teke et al. [95]) to the exciton bound to neutral donors and are labeled as D^0^X/I_4_, D^0^X/I_2,_ and D^0^X/I_9_, respectively. These notations come from the early characterization of bulk ZnO, where 12 transitions labeled from I_0_ to I_11_, were identified and form the basic set of bound exciton transitions characteristic of ZnO [90]. For the shoulder presented in 3.353 eV, Qiu associates an exciton bound to a neutral acceptor denoted as A^0^X [95]. For the shoulder in 3.342 eV, the authors associate a two-electron satellite transition (TES) [52]. Such recombinations are commonly found in the 3.32–3.34 eV spectral region and involve the radiative recombination of an exciton bound to a neutral donor, leaving the donor in the excited state (2s, 2p states). The TES transitions are red-shifted with respect to the DBE energy by the difference between the first excited (2s, 2p) and ground states (1s) of the donor [96]. An identification of the different species in the material can be made to determine the donor binding energies by using the energy difference between the ground state of the donor and relative excited TES states [97]. Finally, the last pair of peaks at 3.303 eV and 3.228 eV were identified as a donor-acceptor pair transitions (DAP) and first-order longitudinal optical (LO) replica (DAP-LO), respectively. The shift between these two peaks is near the theoretical LO phonon energy in ZnO (70–75 meV) [98]. The broadening of the NBE peak as the temperature is increased (Figure 4) is attributed to the thermal decomposition and ionization of bound excitons as well as to the strong coupling of phonons and excitons [66]. In the work of Cao [99], a low-temperature study of the NBE band of ZnO nanowires is presented. As in the work of Qiu, the photoluminescence intensity increases as the temperature is lowered to 4.2K as a consequence of temperature quenching effects [100]. The main peak they observed at low temperatures was blue-shifted from 3.281 to follow the type of tendency explained by Varshni relationship [101]. The phonon optical replicas (longitudinal LO and transversal TO) were also presented in the excitonic spectra and, as in the work of Qiu, also coincide with the theoretical phonon energies for bulk ZnO [98,102,103,104,105].

A time-resolved study of the NBE photoluminescence of ZnO nanowires with different sizes is presented in the work of Reparaz et al. [94], Figure 5.

They measured the lifetime of the neutral bound exciton at 3.365 eV as a function of the surface/volume ratio of the nanowires and demonstrated that the lifetime decreases for nanowires with smaller diameters. They assume that the systemic increase in the lifetime of the exciton as the nanowire diameter increases is due to surface-induced recombination, i.e., the surface proximity affects the lifetime of the D^0^X exciton. Also, the intensities and recombination dynamics of two broadband transitions in 3.341 (S1) and 3.314(S2) eV were studied. The increasing intensity of the I(S1)/I(D^0^X) and I(S2)/I(D^0^X) ratios when the nanowire diameter is decreased supports the hypothesis of surface states as the origin of these transitions. Also, free-electron to bound-hole transition (e, A^0^) has been assigned to transition S2 related to the intersection of basal plane stacking faults on the material’s surface [106].

#### 3.2.2. Deep Level Emission (DLE)

As mentioned before, the photoluminescence (PL) of ZnO nanostructures exhibits a near-band-edge (NBE) and visible luminescence usually associated with exciton transitions and defect emissions, respectively, Figure 6a [107,108,109,110,111,112]. Several photoluminescence emission centers in the visible region are dependent on the synthesis technique and hence, on the vacancies, surface defects, and morphology of NWs [36]. However, the clear vision of the ZnO visible PL mechanism is still far from being fully understood [113]. The emissions in the region from 3.1 eV down to 1.653 eV are commonly referred to as deep-level emissions (DLE), Figure 6b. The deep levels denote the allowed levels inside the bandgap of the semiconductor that produces transitions with energy in the visible range of the spectrum. The band broadness presumably comes from a superposition of many different deep levels (yellow peak, green peak, and blue peak; Figure 7) that emit at the same time [114]. The green and yellow peaks have been observed in some ZnO systems [115,116,117,118], and the blue peak has been presented in thin films and nanostructured systems [119,120,121]. Some previous work attributed the green and orange luminescence to extrinsic impurities such as Cu, Li, or Fe [122]. Nonetheless, new research shows that undoped ZnO also presents photoluminescence peaks in the visible region and is generally ascribed to the electronic transition from single ionized V_O_^+^ centers to the valence band edge [123,124].

As many reports attribute the origin of these emissions to the presence of defects in the ZnO structure, a brief review of the most common defects in the ZnO lattice must be introduced. Defects in the ZnO structure can be divided into extrinsic and intrinsic levels [49]. The intrinsic, sometimes called native levels, are oxygen vacancy (*V*_O_), zinc vacancy (*V*_Zn_), oxygen interstitial (*O*_i_), zinc interstitial (Zn_i_), oxygen anti-site (*O*_Zn_), and zinc anti-site (Zn_O_) (Figure 6a) [125]. These native defects can exist in different charged states or a neutral state and may form clusters with another intrinsic defect as well as with extrinsic elements. 

One example is the V_O_Zn_i_ cluster, which is formed by an oxygen vacancy and zinc interstitial and is situated 2.16 eV below the conduction band minimum [136]. Also, these native defects are the most common explanation for the fact that ZnO always exhibits a high n-type conductivity [137]. Therefore, minority carrier lifetime, doping, compensation, and luminescence efficiency are frequently directly or indirectly determined by native point defects in semiconductors [138]. However, the reported position of these point defects inside the bandgap often varies [53]. A summary of the position of various energy levels inside the bandgap of ZnO due to intrinsic/extrinsic point defects in the lattice is presented in the work of Willander et al. [114].

Additionally, different widely reported levels are also summarized in this work in Figure 8. In order to understand the contribution of the different point defects in the photoluminescence, the formation process of these luminescence centers (traps) has been represented by different color codes in Figure 8 for the easy understanding of the readers. 

According to Janotti et al. [107] at thermodynamic equilibrium and in no defect-defect interaction regime, the concentration of a point defect (*c*) depends on its formation energy as shown in the following equation: *c* = *N*_sites_(−*E*^f^/*k*_B_*T*)
where *N*_sites_ represents the number of sites where the defect can be incorporated, *k_B_* is the Boltzmann constant, *T* is the temperature, and E^f^ is the formation energy of the specific point defect. It is essential to point out that the previous equation neglects the volume and formation of entropy contributions to the defect concentration that tends to become important at high pressures and temperatures. However, this equation shows that defects with high formation energies will be presented in lower concentrations as higher energy needs to be consumed in their formation. 

The formation energy *E*^f^ of point defects in the ZnO lattice depends on the environment, which can be zinc-rich, oxygen-rich, or somewhere in between. Therefore, the abundance of oxygen and zinc atoms in the growth environment controls the concentration of point defects in ZnO. The formation of energy can also depend on the chemical potential (μ) if the defect state has an associate charge [139]. 

It is also important to note that the formation of energy is also a function of the morphology, as defect formation energies at the surface are lower than in bulk, resulting in high defect concentrations at the surface. As mentioned before, one of the characteristics of ZnO nanostructures is that they usually have a substantial surface-volume ratio. Then, the surface effects become dominant in the photoluminescence of ZnO nanowires [113].

Another surface effect is the absorption of gas or determined biomolecules on the surface of ZnO nanowires, which can strongly influence in photoluminescence properties of nanowires [2,140,141,142,143]. The earlier mentioned principle is employed for the use of ZnO nanowires as optical biosensor layers and gas detectors transducers [73,144]. The smaller the diameter of the nanowire, the larger surface-volume ratios it exhibits and the more substantial surface effects it can display [110]. While it is clear that the surface effects and surface band bending are important to ZnO-visible PL, the mechanism of how the surface effects influence the photoluminescence is still debatable [113].

#### 3.2.3. Blue Photoluminescence Band (BPL)

The blue emission is commonly attributed in the literature to the transition between CB and zinc vacancies [145] or oxygen antisites *O*_Zn_ [146]. Oxygen antisites can be produced by irradiation or ion implantation to reach non-equilibrium conditions [138]. A brief summary of the band levels has been presented further in Figure 10.

**Figure 8 nanomaterials-10-00857-f008:**
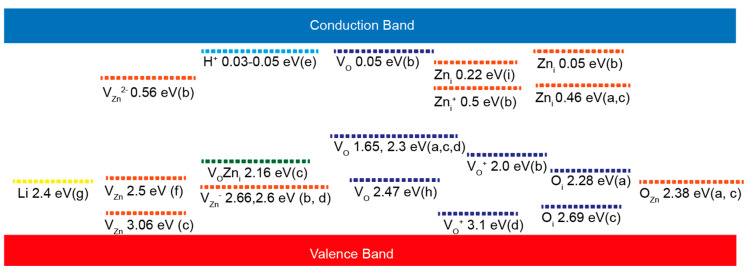
A summary of different defect-related energy levels in the ZnO bandgap reported in the literature by different groups: (**a**) [129]. (**b**). [145]. (**c**). [133]. (**d**). [147]. (**e**). [148]. (**f**). [138] (**g**) [149]. (**h**) [146]. and (**i**). [150]. A color code has been used to point out the different defect states, e.g., the Zn defect is represented by orange lines, oxygen defects are presented by a dark violet color, the Li defect is presented as a yellow bar, the hydrogen level is represented by a light-blue bar, and the complex state attributed to the combination of oxygen and zinc defect states is represented by a green color.

#### 3.2.4. Green Photoluminescence Band (GPB)

For decades, the origin of the green photoluminescence band (2.4–2.5 eV) for undoped ZnO remained divisive [114]. As previously mentioned, most of the intrinsic point defects create different energy levels inside the bandgap of the semiconductor, and therefore, different photoluminescence emissions along the visible range are observed. While some authors attribute the origin of the green band due to the transitions from deep donor levels of oxygen vacancies (*V*_O_) [124,151] and interstitial Zn atoms (Zn_i_) [118,152] to the valence band, it could also be due to the transition from a shallow donor state to zinc vacancies (*V*_Zn_) [114]. In addition to these intrinsic point defects, some works also report the presence of impurities (mainly copper [153,154]) as responsible for the green emission of ZnO. 

From the above-mentioned defects, the dominant native defects are *V*_O_ or Zn_i_ since the growth of ZnO nanowires is usually carried out in a Zn-rich environment. The unintentional intrinsic n-type doping and conductivity are also ascribed to these native defects [155]. Since 1954 [156], V_O_ has been the most widely reported defect responsible for the photoluminescent center in the green region. However, according to some of the theoretical works, such as that presented by Janotti et al. [138], the formation energy of *V*_O_ is high (around 3.72 eV) in zinc-rich growth conditions (*n*-type ZnO), and according to the concentration of the point defect (c) equation presented earlier, their amount would not be significant.

Then, as the formation energy of V_Zn_ is the lowest of all the intrinsic defects in *n*- ZnO [138], some authors have proposed models taking this defect as the source of the green luminescence [70,157]. Zinc vacancies usually introduce bandgap states that are partially occupied states and are formed from broken oxygen bonds that introduce levels in the bandgap that are partially occupied and act as deep acceptors 0.9 eV above the valence band [119,138], Figure 8. The transition from a shallow donor or the conduction band edge to these zinc vacancies level would produce photoluminescence around 2.4 eV (green color) [114].

Corroborations of both oxygen vacancy and zinc vacancy models have been presented in the work of Fabbri. et al. [70] and Kennedy [76]. In these works, the spatial distribution of NBE of ZnO nanostructures by scanning electron microscopy cathodoluminescence techniques (SEM-CL) (Figure 9) is presented. Even though these authors also attribute the green emission to distinct point defects ([70] to *V*_Zn_ and [76] *V*_O_, there is a compromise that the defect responsible for the green emission is mainly located at the surface [136]. The layer responsible for the green emission was estimated to be 30 nm thick (Hsu et al. [158]) in polarized photoluminescent measurements of aligned ZnO nanorods. It has also been shown that coating ZnO nanostructures with a surfactant suppresses green emission [159].

These results are consistent with the spatial distribution of green emission centers seen in the SEM-CL images reported in [70,76], where it can be seen that the most intense green emission occurs near the surface of the nanostructures or at grain boundaries.

Finally, the third intrinsic defect proposed to be involved in the green emission is zinc interstitials (Zn_i_). They generate a donor state level 0.22 eV [146,150] below the conduction band, then the energy level may inject electrons from the conduction band to V_O_ [132]. However, their high formation energy (6 eV under Zn-rich conditions) indicates that Zn_i_ will be present in small concentrations and cannot contribute significantly to the green photoluminescence [138].

In addition to the above-mentioned intrinsic defect, other extrinsic deep levels, including Cu, have been studied [160]. The identification of the Cu defect as being responsible for the green band was made by measuring the PL spectrum as a function of the magnetic field [161]. In response to the applied magnetic field, one band (at around 2.85 eV) split into two with its characteristic fine structure due to the presence of CuZn defect. Also, the measured g-factors from the angle dependence of the splitting are nearly identical to those of a divalent copper ion Cu^2+^ in a zinc lattice site of ZnO measured by electron paramagnetic resonance (EPR) [161]. A representative scheme of these transition states is presented in Figure 10a–c. 

#### 3.2.5. Yellow Photoluminescence Band (YPB)

The yellow photoluminescence band has been ascribed to O_i_ defects [162,163] but most commonly to the lithium doping effect in the ZnO electronic structure [53,149,164]. Li generates states that act as deep acceptor levels, 0.8 eV over the valence band. The transition from the conduction band to these generated levels inside the bandgap leads to a yellowish emission. In the work of Zwingel [164], ZnO samples doped with lithium showed a yellow photoluminescence band centered at 2.2 eV measured from 4.2 K to 300 K. 

He also reports a slower decay in the intensity of the yellow photoluminescence band (YPB) in comparison to the green band (GPB) after turning off the excitation laser. The temperature dependence studies of the GPB and YPB suggest that DAP-type transitions from the conduction band or shallow donor to the Li acceptor dominate at low temperatures [165]. It can be seen in ZnO: Li that YPB dominates over the GPB at low temperatures but is quenched at temperatures above 200 K. 

On the other hand, the common proposed intrinsic defect responsible for the YPB is also the interstitial oxygen sites (O_i_ ) [129,166]. The position of the O_i_ level is approximately 2.28 eV below the conduction band, so the transition between the conduction band and this level will produce the yellow color. Orange emission involving the same O_i_ levels and zinc intersticial levels located 0.22 eV below the conduction band have been reported, so the transition between the levels of Zn_i_ to O_i_ is estimated to be approximately 2.06 eV [129,134,150], Figure 8 and Figure 10d–f. 

The yellow photoluminescence band has been observed in chemically prepared hexagonal ZnO nanorods and is attributed to O_i_ defects [167] and Zn(OH)_2_ groups which could leave a residue of the growth technique on the surface of the nanostructures. It was later shown that the yellow emission could be substituted for the green and red bands by a heat-treatment desorption process of the hydroxyl groups [167]. 

**Figure 10 nanomaterials-10-00857-f010:**
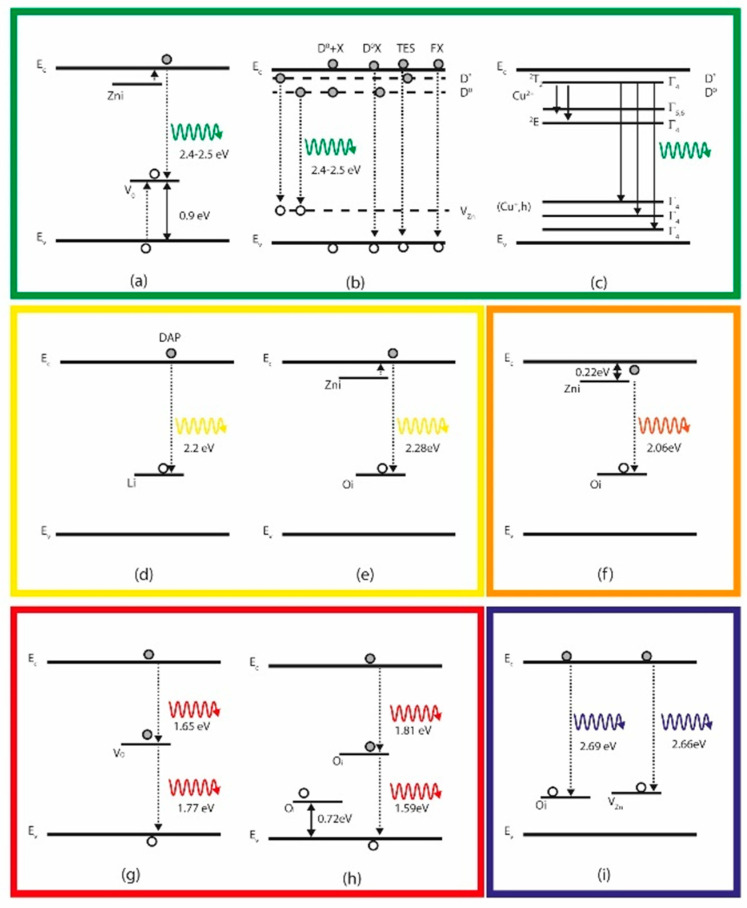
Proposed deep level transition in the different colors indicating the defect involved (**a**) [132], (**b**) [157], (**c**) [154], (**d**) [165], (**e**) [166], (**f**) [134], (**g**) [130], (**h**) [168,169], (**i**) [145,146,149].

#### 3.2.6. Red Photoluminescence Band (RPB)

The photoluminescence emission of ZnO in the red region of the spectrum has been reported in the work of Özgür [63]. The band is centered at 1.75 eV and has an FWHM of 0.5 eV. Similar to the yellow emission, the red band also is quenched if the system is heated from 15 to 300 K, giving rise to the green band, which could subsequently dominate the photoluminescence spectra. The earlier discussion reflects the possible competition for recombination between the acceptors of the green and red bands. In the work of Alvi et al., oxygen vacancies situated 1.65 eV below the conduction band are considered responsible for the red emission of ZnO nanotubes [130], Figure 10g,h.

On the contrary, under oxygen-rich conditions, the ZnO structure can contain oxygen interstitial O_i_ that introduces acceptor states 0.72 and 1.59 eV above the valence band [138] that in theory can produce the transition in the orange-red region [169]. The emission in these regions has also been attributed to zinc vacancy complexes in the work of Gomi et al. [170] and with interstitial zinc atoms in the work of Djurisic et al. [171].

## 4. Summary

Despite the extensive literature published about zinc oxide nanostructures, as far as we know, there are no reviews centered on the discussion related to the emission mechanism in ZnO nanowires. These characteristic properties make ZnO an exciting candidate for many different practical applications such as biological, energy, and optoelectronic devices. 

The effect of a size reduction over the main bands of emission has been discussed in this review. Furthermore, a brief summary of different native point defect states or trap centers has been presented in a simple tabular and schematic form for an easy and better understanding of the topic.

## Figures and Tables

**Figure 1 nanomaterials-10-00857-f001:**
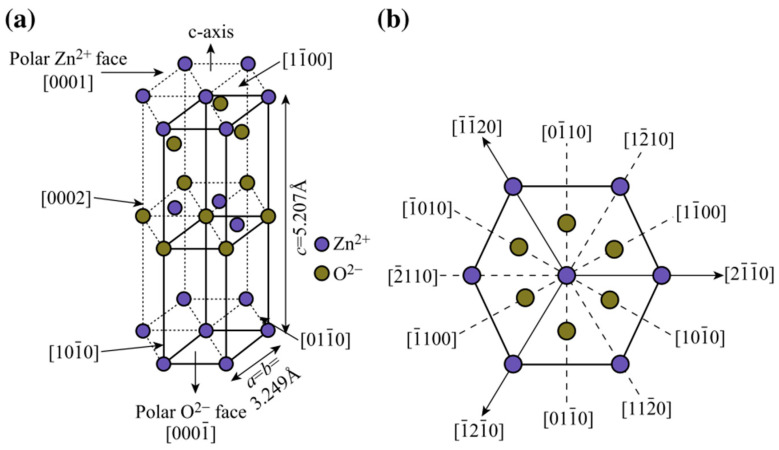
(**a**) ZnO unit cell with wurtzite structure. (**b**) Various crystal planes of the ZnO wurtzite structure [42].

**Figure 4 nanomaterials-10-00857-f004:**
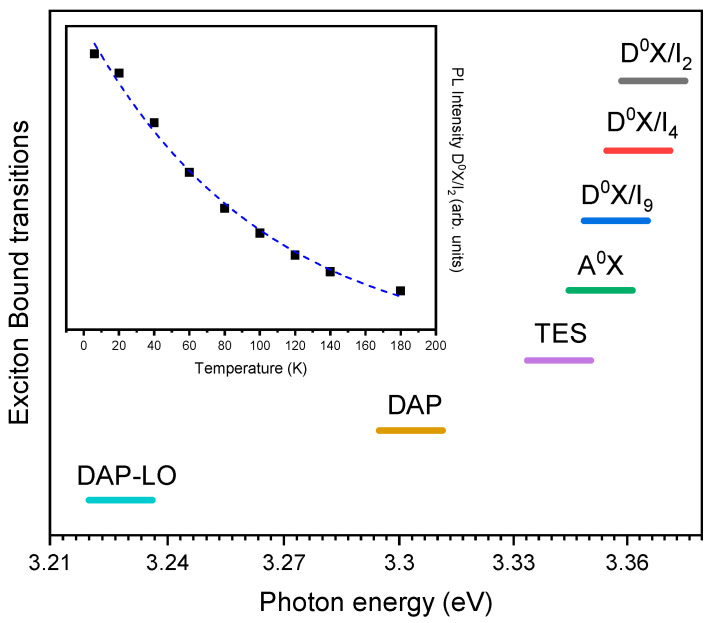
The exciton-bound transition energy reported in [66] at 6 K. D^0^X (denotes exciton bound to neutral donors), A^0^X (exciton bounded to a neutral acceptor), TES (two-electron satellite transition), DAP (donor-acceptor pair transitions), and DAP-LO (first-order longitudinal optical). Quenching effect of the D^0^X/I_2_ emission as the temperature is increased from 6 to 180 K.

**Figure 5 nanomaterials-10-00857-f005:**
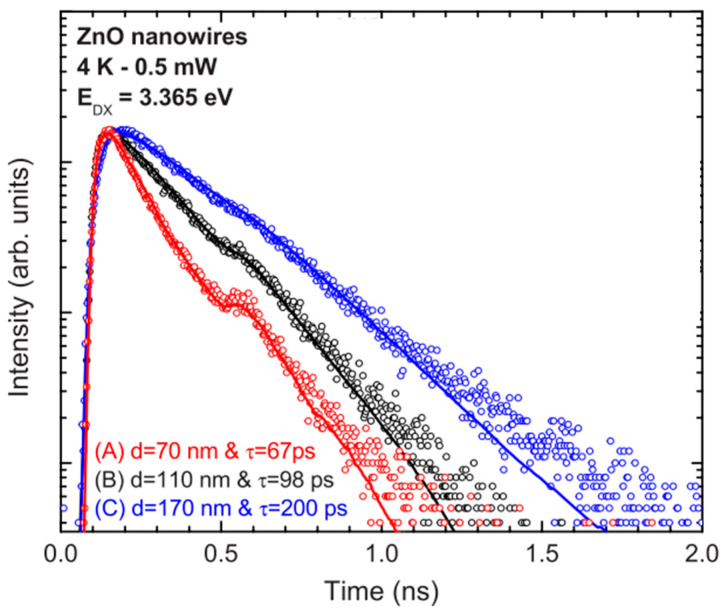
Low-temperature time-resolved photoluminescence (TRPL) spectra at 3.365 eV for three ZnO NWs arrays with different diameters. A diameter dependence over the exciton lifetime was observed [94]. Copyright 2010 AIP Publishing.

**Figure 6 nanomaterials-10-00857-f006:**
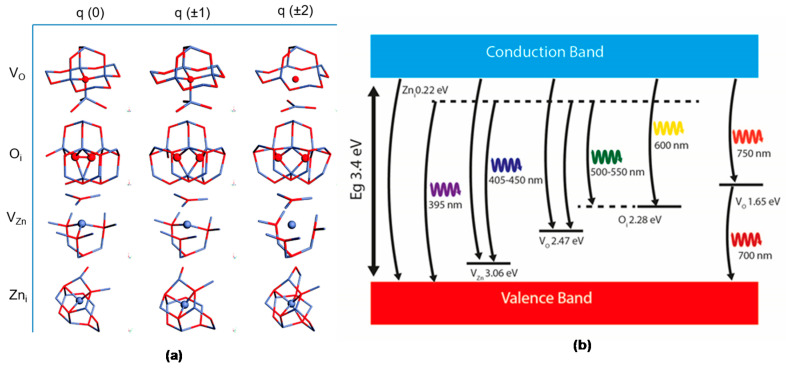
(**a**) Illustration of different intrinsic point defects in the ZnO lattice. Oxygen vacancy (*V*_O_), zinc vacancy (*V*_Zn_), oxygen interstitial (*O*_i_), and zinc interstitial (Zn_i_) in different charge configurations are presented [125]. Copyright 2006 Royal Society of Chemistry (**b**) Schematic band diagram of the DLE emissions in ZnO based on the full-potential linear muffin-tin orbital method and the reported data as described in references [126,127,128,129,130,131,132,133,134,135]. Also, oxygen vacancies situated 1.65 eV below the conduction band are denoted to contribute to the red emission [130].

**Figure 7 nanomaterials-10-00857-f007:**
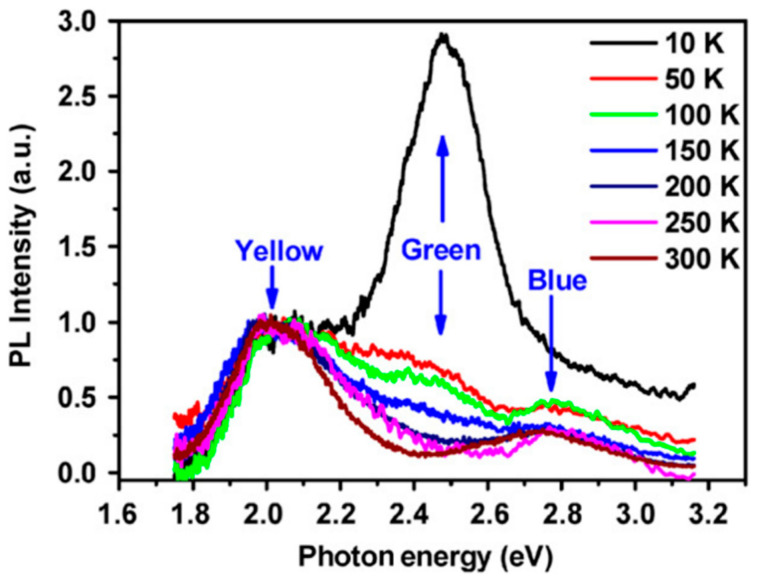
Visible PL spectra of a single ZnO nanowire measured at different temperatures. Three characteristic emission bands (yellow, green, and blue) were observed [113]. Copyright 2008 Elsevier.

**Figure 9 nanomaterials-10-00857-f009:**
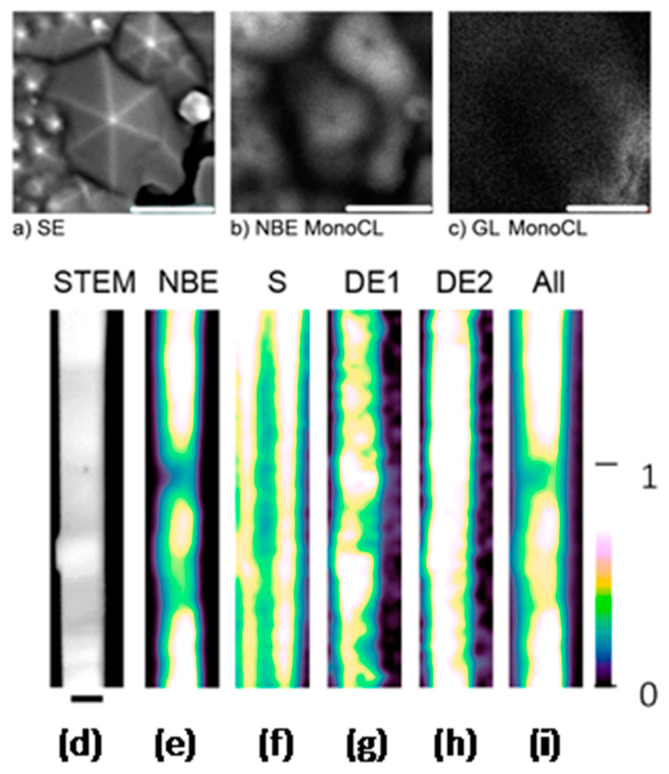
(**a**) Secondary electron micrograph of the ZnO thin film, (**b**) CL monochromatic map taken under ZnO NBE emission, (**c**) Green emission CL monochromatic map at ZnO; all the scale bars are 500 nm long [70]. (**d**) STEM image of nanowire amplitude maps of (**e**) near-band-edge emission, (**f**) surface emission, (**g**) defect emission 1, and (**h**) defect emission 2 deconvolved components. (**i**) Counts integrated across all energies; for further permissions related to the material (d–i), please refer to ACS https://pubs.acs.org/doi/10.1021/acs.jpclett.8b03286 [76].

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
