# Peer review of "Photoluminescence of ZnO Nanowires: A Review"

_nanomaterials, 2020, doi:10.3390/nano10050857_

Round 1

Reviewer 1 Report

This manuscript provides an overview of the photoluminescence properties of ZnO nanowires. The article is well structured with an excellent introduction and the references are adequate. This paper is suitable for the special issue " Growh and Characterization in nanowires" in Nanomaterials journal.

Author Response

We thank the reviewer for his views for the manuscript entitled   “Photoluminescence of ZnO nanowires: A review”.

Reviewer 2 Report

This paper is well ordered about the luminescence properties of ZnO nanowires. Related researcher give some information about the structure and characterization. But some correction parts were points out as follows.

  1. Wirte the title of 2 and 3.
  2. In the Table 1, check out the position of reference [73].

Author Response

We thank the reviewer for his views for the manuscript entitled “Photoluminescence of ZnO nanowires: A review”. Please find all of the changes marked with yellow in the document attached.

  1. Wirte the title of 2 and 3.

Reply: Thank you for the comment. We have again described in a better and clear way the title of Fig. 2 and 3.

2. In the Table 1, check out the position of reference [73].

Reply: Thank you for the excellent observation. We have cross-checked the position of reference 73 in the manuscript. In Reference 61 and 73, we have shown the formation and explained different properties, especially PL of nanowires. Also, in both these articles, we have discussed briefly the mechanism of PL. However, as suggested now, we have cited both these references in Table 1 to have a detailed and better review of the PL properties of these kinds of nanowires formed by the VLS technique ( work of our group).

Reviewer 3 Report

Good and thorough review paper with an extensive reference list, indicating that the field has been well covered.

Figures like Figure 10 using photoluminescence color coding very instructive.

However, there are some shortcomings:

  1. Confusing different notations of ZnO crystal planes, for instance in line 105 using (11-20) while in Figure 1 [1120] is used. The same is seen in Line 316 for transitions with (LO-DAP) while figure 4 uses (DAP-LO). Several other places also. Also donor notations in line 304 is different from what is used in Figure 4. Please harmonise all through the manuscript.
  2. Acronyms should be speeled out the first time they are used. 0D, 1D, etc.
  3. Line 115: archived? achieved?
  4. Line 126: obtention?
  5. You are using nonscientific subjective verbs, adjectives, adverbs, etc. such as "little" (Line 111), "most probably" (Line 308), "suggest" (L338), "possibly" (L345), etc. Please scientific terms that these are yet unproven hypotheses. If not corrected, the manuscript should be rejected, or else the readers might be misguided.
  6. Overall, figure captions and legends should explain the figures clearly without having to consult the the text elsewhere.
  7. Line 385: Photon, not photo.

Author Response

We thank the reviewer for his views for the manuscript entitled “Photoluminescence of ZnO nanowires: A review”. Please find all of the changes marked with yellow in the document attached.

  1. Confusing different notations of ZnO crystal planes, for instance in line 105 using (11-20) while in Figure 1 [1120] is used. The same is seen in Line 316 for transitions with (LO-DAP) while figure 4 uses (DAP-LO). Several other places also. Also donor notations in line 304 is different from what is used in Figure 4. Please harmonise all through the manuscript.

Reply: Thank you for the excellent observation.We have harmonized the terminology in the manuscript and cross-checked. Please find the amendments made highlighted with yellow color in the manuscript.

2. Acronyms should be speeled out the first time they are used. 0D, 1D, etc.

Reply: Thank you for the observation.We have tried to spell all of the acronyms in the first place they are cited in the manuscript. Please find the modification made highlighted with yellow color.

3. Line 115: archived? achieved?

Reply: Thank you for the observation. There was some writing error in this line. Please find the correction made  highlighted with yellow color.

4. Line 126: obtention?

Reply: Thank you for the observation. Please find the correction made  highlighted with yellow color.

5. You are using nonscientific subjective verbs, adjectives, adverbs, etc. such as "little" (Line 111), "most probably" (Line 308), "suggest" (L338), "possibly" (L345), etc. Please scientific terms that these are yet unproven hypotheses. If not corrected, the manuscript should be rejected, or else the readers might be misguided.

Reply: Thank you for the excellent observation and recommendation. We have tried to improve the way of writing the adverbs. Please find the corrected and modified scientific adverbs with yellow color in the manuscript.

6.Overall, figure captions and legends should explain the figures clearly without having to consult the the text elsewhere.

Reply: Thank you for the recommendation. We have re-written the figure legends in a descriptive way for a better understanding of the readers.

7. Line 385: Photon, not photo.

Reply: Thank you for the observation. Please find the correction made  highlighted with yellow color.